# Finite Element Analysis on Ultrasonic Drawing Process of Fine Titanium Wire

**Shen Liu** **, Xiaobiao Shan, Hengqiang Cao and Tao Xie ***

School of Mechatronics Engineering, Harbin Institute of Technology, Harbin 150001, China;
joseliu2013@outlook.com (S.L.); shanxiaobiao@hit.edu.cn (X.S.); yzhinv@outlook.com (H.C.)
* Correspondence: xietao@hit.edu.cn; Tel.: +86-451-8641-7891; Fax: +86-451-8641-6119

**Abstract:** Ultrasonic drawing is a new technology to reduce the cross-section of a metallic tube, wire or rod by pulling through vibrating dies. The addition of ultrasound is beneficial for reducing the drawing force and enhancing the surface finish of the drawn wire, but the underlying mechanism has not been fully understood. In this paper, an axisymmetric finite element model of the single-pass ultrasonic drawing was established in commercial FEM software based on actual wire length. The multi-linear kinematic hardening (MKINH) model was used to define the elastic and plastic characteristics of titanium. Influences of ultrasonic vibration on the drawing process were investigated in terms of four factors: location of the die, ultrasonic amplitude, drawing velocity, and friction coefficient within the wire-die contact zone. Mises stresses, as well as contact and friction stress, in conventional and ultrasonic drawing conditions, were compared. The results show that larger ultrasonic amplitude and lower drawing velocity contribute to greater drawing force reduction, which agrees with former research. However, their effectiveness is further influenced by the location of the die. When ultrasonic amplitude and drawing speed remain unchanged, the drawing force is minimized when the die locates at the half-wavelength position, while maximized at the quarter-wavelength position.

**Keywords:** ultrasonic drawing; titanium wire; drawing force; finite element analysis; Mises stress; contact stress; work hardening; numerical simulation

---

## 1. Introduction

Titanium wires are widely used in various industries, including aerospace, automobile, biomedicine, petrochemistry, fishery, due to their exceptional characteristics, such as high strength, lightweight, good corrosion resistance, excellent biocompatibility, etc. [1–4]. However, unlike many other metallic wires, the manufacturing of titanium wire is usually conducted at elevated temperature, as their cold workability is degraded by high yield stress to tensile strength (Y/T) ratio and the strain hardening phenomenon [5]. The conventional drawing process involves complicated heat preparation, lubrication, and surface treatment, which is neither eco-friendly nor energy-saving. What is more, the wire products often have low dimensional accuracy, poor surface finish, and high breakage ratio [6]. Ultrasonic drawing is a metalworking process to reduce the diameter of a wire, tube, or rod by pulling them through oscillating dies with converging cross-section shapes [7]. Compared with the conventional drawing process, the new technology has the following advantages: reduced drawing force, enhanced surface finish, lower breakage ratio, prolonged tool life, greater area reduction per pass, etc. Therefore, ultrasonic drawing is a promising technology to increase productivity, lower costs, and improve the quality of the products in the industrial manufacturing of metallic wires [8]. The benefits associated with ultrasound are not limited to titanium and the drawing process, but to varied machining processes, such as welding, forging, rolling, and cutting of a wide range of metallic and non-metallic materials like aluminum, steel, and carbon fiber [9–11].

The idea of introducing ultrasonic vibrations into the drawing process was first reported by Blaha et al. in 1955 [12]. They discovered the acoustic softening phenomenon, which is similar to thermal softening effects, with an abrupt 40% reduction in drawing force when ultrasound was superimposed. The observations were explained as the result of activation and increased mobility of dislocations at the lattice defects. In 1968, Winsper et al. reviewed various experimental results on ultrasonic-assisted metal forming processes and pointed out that all these phenomena could be explained by the superposition of an alternating load on a static load, which is later known as stress superposition hypothesis [13]. The above theories triggered numerous studies on the influence of ultrasound in metal forming processes, which remains active to the present day. In 2003, Hayashi et al. compared the influence of longitudinal and radial ultrasonic vibrations on the aluminum wire drawing process using the finite element method (FEM) [14]. It was found that radial vibration yielded superior results than longitudinal vibration with respect to drawing force reduction, surface quality improvement, and critical drawing speed. In 1999, Susan et al. conducted the ultrasonic drawing experiment of steel ball-bearing steel wire. They attributed the drawing force reduction to the surface effect of ultrasound utilizing the friction reversion mechanism [15]. The friction coefficient and the drawing force were calculated according to the relations between three factors: drawing velocity, ultrasonic frequency, and amplitude of the die. In 2004, the proposed calculation method was further revised to calculate the drawing force in the ultrasonic-assisted tube drawing process [16]. They also pointed out that ultrasonic vibration contributes to weakening the strain hardening effect and helps to increase the plasticity of the specimen. In 2009, Qi et al. experimentally examined the effects of longitudinal ultrasonic vibrations in the industrial production of brass wire [17]. Results proved the advantages of ultrasonic drawing over the conventional drawing with a 17% increment in drawing speed, 7% decrease in drawing force, prolonged tension regulation period from 0.5 s to about 1.5 s, and improved surface finish of the drawn wire. In 2012, Shan et al. proposed a new mathematic model to describe the ultrasonic drawing process based on the nonlocal friction theorem and obtained the stress distribution, contact pressure, and drawing force [18]. The ultrasonic anti-friction effect of stainless-steel wire, copper wire, and titanium wires at room temperature were analyzed. In 2016, Yang et al. compared the influences of longitudinal-torsional composite vibration and longitudinal vibration on the titanium wire drawing process through numerical and experimental methods [19]. The results indicated that longitudinal vibration is more beneficial for reducing the friction force and improving the surface finish of the drawn wire than composite vibration. In 2018, Liu et al. performed an experimental study on two-pass titanium wire drawing with two oscillating dies and achieved a drawing force reduction of over 50% [20]. Results showed that increased vibration amplitude leads to greater drawing force decrement, whereas increased drawing velocity brings about the reverse effect. Higher drawing speed with moderate ultrasonic amplitude is more preferable in removing surface defects.

Although the feasibility and advantages of ultrasonic wire drawing have been verified by previous researchers, the underlying mechanisms were explained in different ways. The correctness of presuppositions in quantitative calculations remains to be verified. The recent works mainly focused on experimental research and theoretical studies, whereas little attention was paid to the finite element analysis of the ultrasonic-assisted drawing process. In this paper, the ultrasonic drawing process was described as contact and friction problems between the elastic-plastic traveling string and the rigid vibrating die. The axisymmetric finite element (FE) model was established in commercial FEM software Abaqus, considering the actual length of the wire and the strain hardening characteristics of the material. Influences of ultrasound on the drawing force were discussed and compared in terms of four factors: vibration amplitude, drawing speed, location of the die, and the friction coefficient. Mises stress distribution inside the wire, as well as contact and friction stress within the contact region, was investigated.

## 2. Finite Element (FE) Model Establishment and Simulation Procedure

The titanium wire drawing process could be explained as the frictional contact problem between a certain length of deformable wire and the internal surfaces of the rigid oscillating die. Considering the inherent symmetry of the geometry and boundary condition, an axisymmetric finite element analysis model was established in commercial software Abaqus (Dassault Systèmes Simulia Corp., Providence, RI, USA) to minimize computational costs, as shown in Figure 1. In this particular problem, the diameter of the raw titanium wire, Ø 0.4 mm, is drawn into Ø 0.36 mm, with a cross-section area reduction of 19%. The generation of heat due to plastic dissipation inside the wire and frictional heat generation within the contact region are not considered, as this paper emphasizes on the independent effects caused by the addition of ultrasound in the general drawing process of long thin metallic wires. The drawn wire is wound up around the drum reel driven by an electrical motor. The length of unwound wire, or the distance between the die and the reel, denoted as viable *L* in the figure, is usually 100~200 mm and should be larger than the radius of the drum, 40 mm, in practice. The flexibility and elastic deformation of slightness titanium wire would exert a remarkable influence on the ultrasonic drawing process, therefore, the uncoiled drawn wire should be modeled with the full length. In contrast, the length of the raw wire, denoted as *l* in the figure, has little influence on simulation results and therefore is assigned as a constant of 10 mm. In the discretization process, titanium wire is defined as a deformable part with an overall meshing size of 0.02 mm, whereas the die is modeled as a rigid body with an element size of 0.03 mm. The two parts are all modeled using the 4-node reduced axisymmetric continuum quadrilateral solid element. Penalty and kinematic formulations are employed in the definition of contact interactions. The contact type between the die and the wire is specified as surface-to-surface contact, and the contact behavior is assumed to abide by Coulomb friction law, with an initial coefficient of 0.1. The simulation is also performed with Arbitrary Lagrangian-Eulerian (ALE) adaptive meshing and enhanced hourglass control.

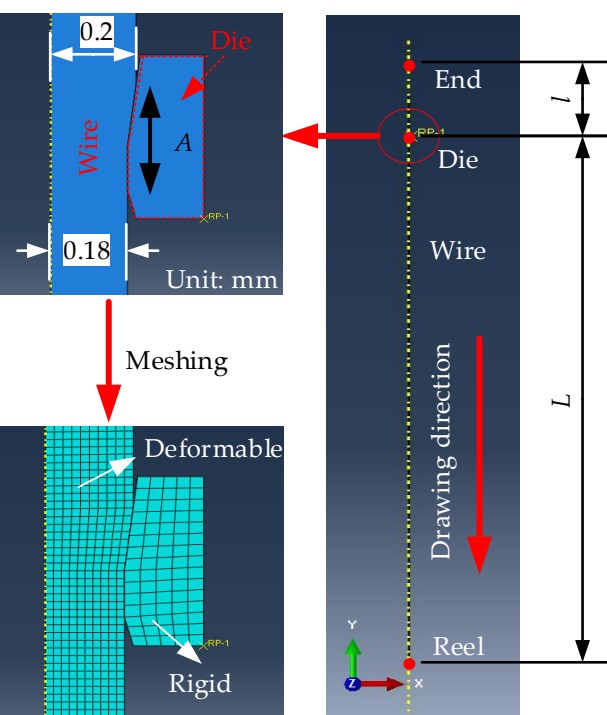

**Figure 1.** Simulation settings and FE model of single-die ultrasonic wire drawing.

In the meshed model, all the nodes situated on the central axis are constrained at the radial (X) and circumferential (Z) direction. Velocity constraints along the wire drawing direction (−Y) are prescribed to the bottom nodes of the wire near the reel. A reference point NREF is assigned to control

the movement of the rigid die. The whole simulation process, which lasts for 5 ms, can be divided into three steps. The duration for the first and the third step is 2 ms, which are separately divided into 100 substeps. In these two steps, the amplitude of the die is assigned as 0 to simulate the conventional drawing process. The second step lasts for 1 ms, which equals to 20 oscillation cycles for, and is divided into 400 substeps, during which ultrasonic vibrations of 20 kHz is applied to the die to stimulate the ultrasonic drawing process. In the study, the frequency of the die remains the same, whereas its amplitude changes from 1 to 10 μm. The variation range of the drawing speed and the distance from the die to the reel are 100~1200 mm/s and 15.8~126.4 mm, respectively. The friction coefficient varies from 0.1 to 0.5. The wire drawing forces could be captured by extracting the reaction force along the Y direction at the NREF point.

In the simulation, the material of wire is selected as TA2 commercial pure titanium, the density, elastic modulus, and Poisson ratio of which are 4500 kg/m³, 115 GPa, and 0.37, respectively. The wire drawing die mainly consists of two sections: a stainless casing and the diamond nib [21]. In the FE model, only the nib section, which directly contacts the wire, is considered. The diamond mandrel, with 3520 kg/m³, 1100 GPa, 0.07, respectively, in density, elastic modulus, and Poisson ratio, could be divided into three areas: the reduction area with the semi-cone angle of 7°, the bearing area, and the exit area. Plastic deformation mainly occurs in the reduction area; however, the bearing area is essential for maintaining the dimensional accuracy of the wire products. The stress-strain curve of TA2, as well as the geometry and dimension of the diamond nib, is shown in Figure 2. The plastic deformation of the titanium is assumed to follow the Mises yield criterion, and a multi-linear kinematic hardening (MKINH) model was employed to describe the plastic behavior of the material [22]. Considering the changing stress state at the wire-die interface caused by the oscillation of the die, the simulation process involves not only material nonlinearity, but also nonlinear geometry and boundary conditions; therefore, the transient analysis is conduct in the Abaqus/Explicit module to improve the efficiency and convergence of the calculation.

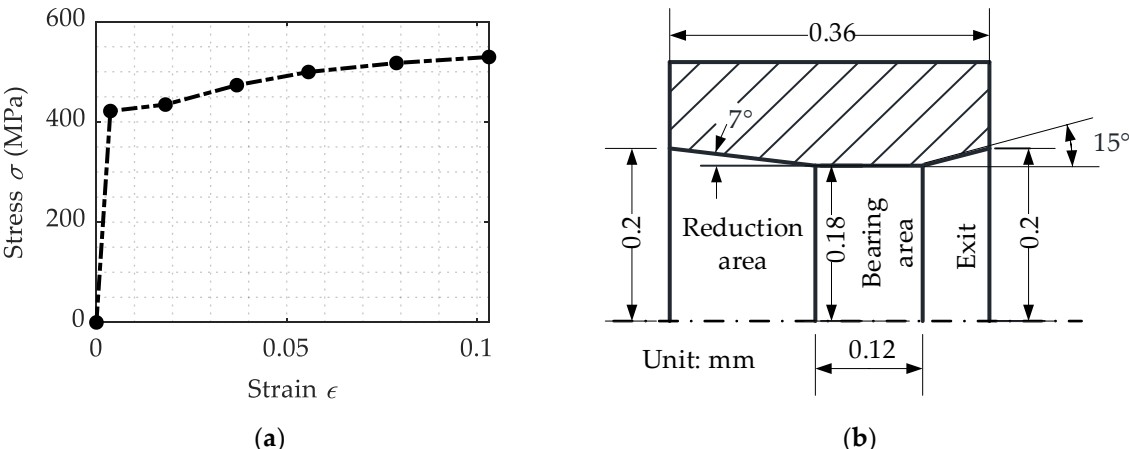

**Figure 2.** Material property and geometry definition of the die specified in the simulation program: (**a**) Stress-strain curve of TA2 commercial pure titanium in FE simulation; (**b**) Structure and dimensions about the diamond nib of the wire drawing die.

## 3. Results and Discussion

Based on the above finite element analysis (FEA) model, influences of ultrasonic vibration on titanium wire drawing process are investigated in terms of four factors: the location of the die, the vibration amplitude of the die, the wire drawing velocity, and the friction coefficient. The relationships between each factor and the averaged drawing force are analyzed using the control variates technique. The internal equivalent stress distribution of titanium wire, as well as the contact

and friction stress distribution at the die-wire interface, is also displayed to understand the underlying mechanism of the ultrasonic drawing process.

### 3.1. Influence of Die Location on Drawing Force

In previous research, Hayashi et al. built an FEA model and discussed the influence of vibration amplitude and direction on drawing force and Mises stress distribution inside the wire when drawn with the assistance of axial and radial ultrasonic vibrations [14]. However, the established model was confined to a very narrow region of 1~2 mm in length adjacent to the die. The flexibility and stretching of the elastic slender wire are neglected. In industrial production, the wire drawing velocity is provided by the reel drum, which rotates at a constant speed. For the conventional drawing process, the velocity of the wire adjacent to the reel drum is equal to that near the die. However, for the ultrasonic wire drawing process, when the die oscillates periodically, they are not equal, as the uncoiled drawn wire would be tightened and relaxed intermittently along with the die. The influence of this neglected factor on the drawing process is determined by the length of uncoiled drawn wire, i.e., the distance between the reel drum and the wire.

To investigate the influence of reel-die distance $L$ on the ultrasonic drawing process, the other factors maintain constant, with ultrasonic amplitude, drawing speed, and friction factor specified as 10 μm, 300 mm/s, and 0.1, respectively. As the drawn wire is forced to vibrate and its natural frequency is determined by the length, simulations are conducted when the wire length amounts to $\lambda/8$, $\lambda/4$, $3\lambda/8$, and $\lambda/2$, respectively, where $\lambda$ denotes the wavelength of ultrasonic vibration when prorogating in the titanium wire. The wavelength can be calculated as 252.8 mm, based on Young's modulus, density of the TA2, and the oscillation frequency of the die, 20 kHz. The variations of drawing force with time under different uncoiled wire lengths are illustrated in Figure 3.

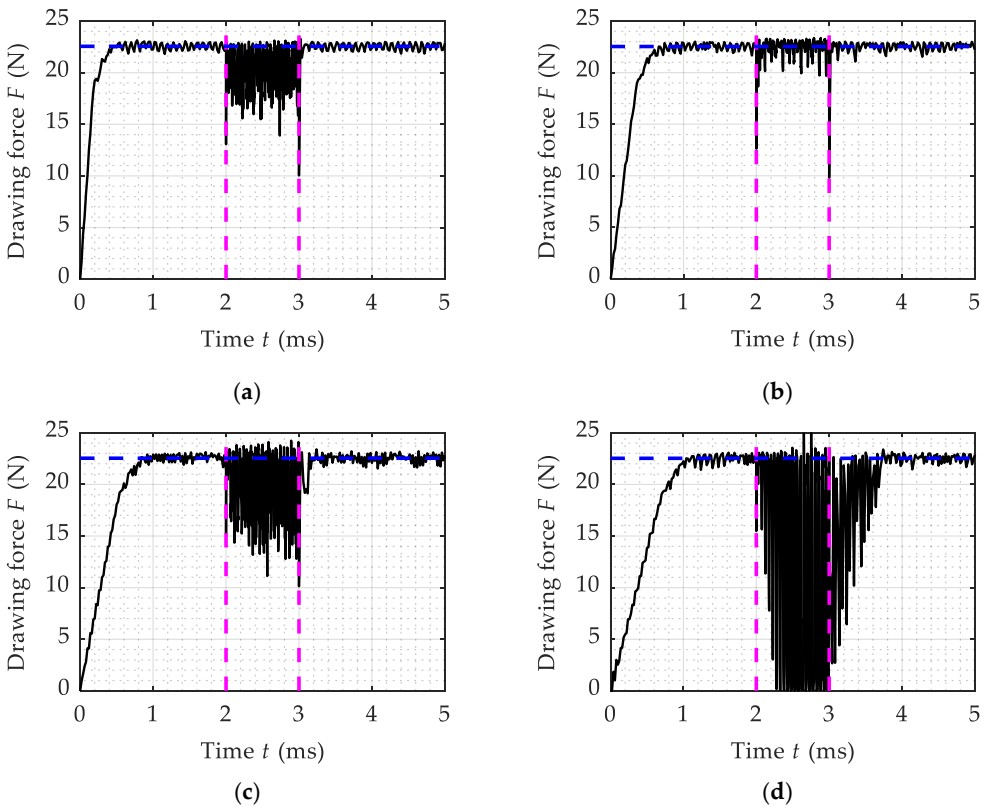

**Figure 3.** Time-variation of the drawing force for different distances between the die and the reel: (**a**) $L = 31.6$ mm; (**b**) $L = 63.2$ mm; (**c**) $L = 94.8$ mm; (**d**) $L = 126.4$ mm.

It can be seen that the whole drawing process lasts for 5 ms. Ultrasonic vibrations were imposed during the period of 2~3 ms and then were removed. At the beginning of the first 2 ms, the drawing force ramped up from 0 N to around 20 N within 1 ms, which represents the elastic deformation phase of the wire drawing process. Afterward, the growth of the drawing force gradually slowed down, which means the start of plastic deformation. The drawing force finally stabilized at around 22.5 N, and wire products were steadily pulled through the die. Comparing Figure 3a,d, we can easily find that the distance between the die and the wire does not influence the conventional drawing force, however, it would affect the duration of the elastic deformation period. When ultrasonic vibrations were applied, the drawing force began to fluctuate in a certain range, with the maximum values slightly larger than or equal to the normal drawing force, whereas the minimum values are significantly lower than that, which results in a decrease in averaged drawing forces. The fluctuation range of ultrasonic drawing force, or the reduction ratio of averaged ultrasonic drawing forces compared to the conventional drawing value, is heavily influenced by the location of the die. Specifically, at the same ultrasonic amplitude and drawing velocity, the drawing force fluctuates most violently when the reel locates 126.4 mm ($\lambda$/2) from the die, while the fluctuation is the minimum when the reel-die distance equals 63.2 mm ($\lambda$/4). Besides, there is a difference between the four subfigures with respect to the envelope curve shapes under the ultrasonic drawing condition. When the die is distributed $\lambda$/2 from the reel, the envelope of the ultrasonic drawing force is shaped like a trapezoid, instead of rectangular when located at other positions. For these positions, the amplitudes of drawing force fluctuation reach the maximum as soon as ultrasounds are imposed, and the drawing forces immediately restored to around 22.5 N when vibrations are removed. At the half-wavelength position, however, the minimum drawing force drops steadily in the first five oscillating cycles before reaching a steady state. When ultrasound is removed, the drawing force remains lower than 22.5 N in about 0.7 ms, instead of going back to the original value right away. Finally, even in the half-wavelength condition and even without the consideration of the pliability of the metallic wire, the ultrasonic drawing forces remains above 0, which means no separations occur between the wire and die at their contact interface, and the validity of the presuppositions for reversed friction mechanism should be further verified. Unlike a bar, a long thin wire could not bear the pressure, which is another reason why the drawing force stays above 0 and the separation could not happen.

Figure 4 shows the variation of averaged drawing force with the distance between the die and the reel. It can be seen that, as the reel-die distance increases from 15.8 mm ($\lambda$/16) to 126.4 mm, the averaged drawing force first goes up and then drops, peaking at the quarter-wavelength position, however, remains below the conventional drawing force value of 22.5 N. At the half-wavelength position, the averaged drawing force reaches the minimum value of 10.28 N, with a reduction of more than 50% compared with conventional drawing force. The influence of the reel-die distance on the ultrasonic drawing force can be attributed to the stretching vibration of the uncoiled drawn wire. This section of titanium wire resonates under the drive of vibrating die when its length approximates half-wavelength, as the frequency of the die is approaching the first-order longitudinal resonant frequency of the wire.

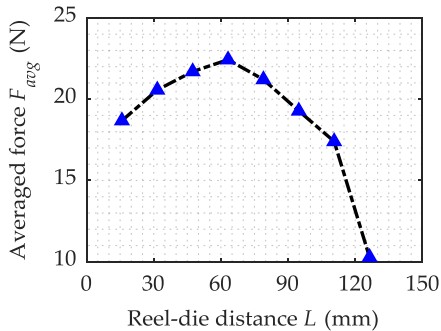

**Figure 4.** FEM calculation results of average ultrasonic drawing forces at different distances between the die and the reel drum.

### 3.2. Influence of Ultrasonic Amplitude on Drawing Force

It can be seen from the above analysis, with the same ultrasonic amplitude and drawing speed, maximum drawing force reduction can be achieved when the distance between the reel and the die approximates half-wavelength of ultrasound traveling in the titanium wire. Therefore, the influences of other factors, including ultrasonic amplitude $A$, wire drawing speed $V$, and friction coefficient $\mu$, on the ultrasonic drawing force will be further considered under this condition. From previous studies, it can be known that the effects of ultrasonic amplitude and drawing speed on ultrasonic drawing force are correlated [14–20]. Drawing force reduction can only be realized when the prerequisite $V < 2\pi fA$ is satisfied.

Figure 5 illustrates the changing of wire drawing force along with time at the drawing velocity of 300 mm/s and with the friction coefficient of 0.1. It can be found that the upper limit of the drawing force remains around 22.5 N, which will not be affected by the intensity of ultrasonic vibrations. In Figure 5a, where ultrasound amplitude is 2 μm, there is almost no change in drawing force compared with the conventional drawing value, because the drawing speed, 300 mm/s, exceeds the threshold value $2\pi fA$, which is determined by the oscillation amplitude and frequency of the die. In other subfigures, where pre-condition is satisfied, an obvious decrease in drawing force can be observed. The envelope lines of ultrasonic drawing force present the shape of the triangle or trapezoid instead of rectangular, as the die locates half-wavelength off the reel drum, which is consistent with previous analysis. The influence of ultrasound will also last for a short time after the die stops vibrating, instead of disappearing instantly. However, the fluctuation range of the ultrasonic drawing force will be widened with the increment of ultrasonic amplitude. When ultrasound is turned on, the minimum values of the drawing force go down consistently until arriving at the stable phase. The time consumption for this period is shortened with larger ultrasonic amplitude, and the decrement of minimum drawing force for each oscillation period will be enlarged correspondingly.

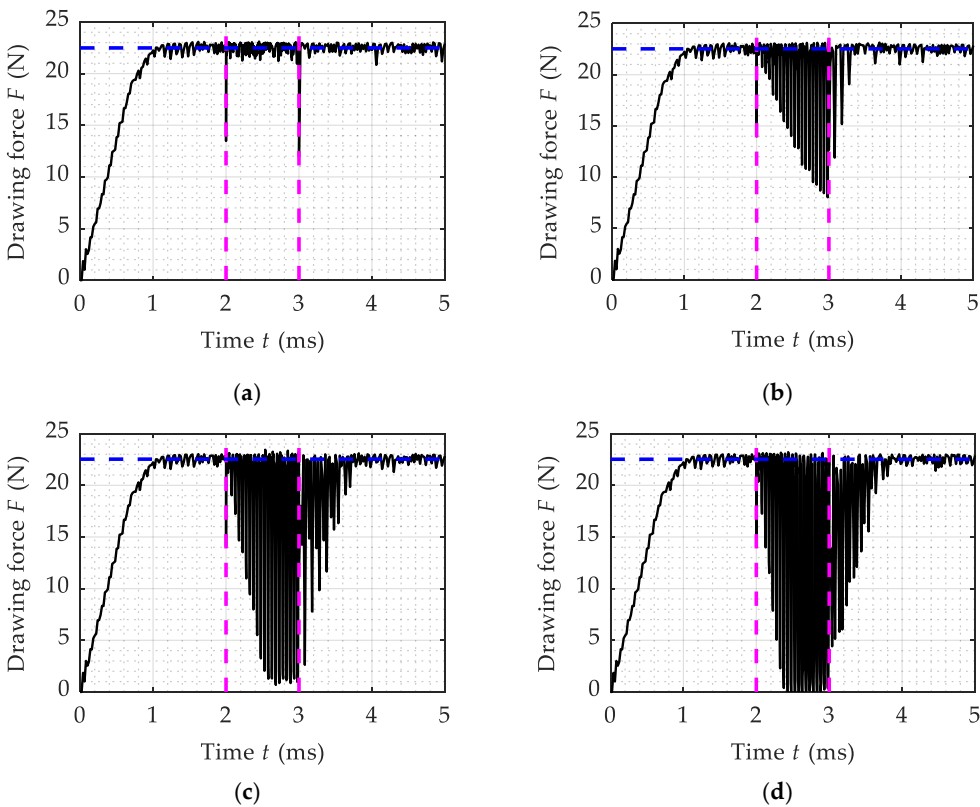

**Figure 5.** Influence of ultrasonic amplitude on time-variation of the drawing force: (**a**) $A = 2$ μm; (**b**) $A = 4$ μm; (**c**) $A = 6$ μm; (**d**) $A = 8$ μm.

Figure 6 shows the numerical calculation results of averaged ultrasonic drawing force under different ultrasonic amplitudes at drawing speed of 300 mm/s and 600 mm/s, respectively. Overall, the two variation curves show the same trend, that is, the averaged ultrasonic drawing force goes down with the increment of ultrasonic amplitude. However, the drawing force declines faster at a relatively lower drawing speed of 300 mm/s. A flat section could be found at the initial segment of the two curves, where there is almost no reduction in drawing force compared with 22.5 N. Therefore, at a certain drawing velocity, to achieve drawing force reduction, the intensity of ultrasonic vibration has to be large enough. The flat region expands at a relatively higher drawing velocity because the threshold drawing speed (V = 2πf$A$) increases with ultrasonic intensity.

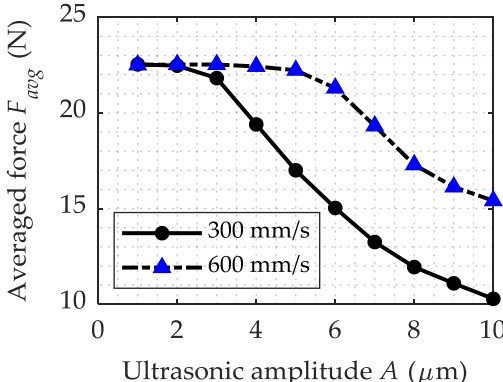

**Figure 6.** FEM calculation results of average ultrasonic drawing force with different ultrasonic amplitudes at drawing speed of 300 mm/s and 600 mm/s.

Based on the above simulation results, the following hypothesis could be put forward. With the addition of ultrasonic vibration, the conventional continuous drawing process is turned into an intermittent ultrasonic drawing process. A vibration cycle of the die can be divided into two phases: the plastic deformation phase, and the stretching vibration phase. For the former, titanium wire is drawn through the die and the drawing force equals the conventional drawing force. For the latter, however, the wire vibrates along with the die. The drawing force is determined by the stretching force of the wire and should be lower than the conventional drawing force. The time allocation between the two phases is determined by both the amplitude of the die and the wire drawing velocity. With the increment of drawing speed, the proportion of the plastic deformation stage will rise correspondingly, therefore, averaged ultrasonic drawing force will be closer to normal drawing force 22.5 N. On the contrary, when ultrasonic amplitude increases, the time duration of the deformation stage is decreased, and the stretching force vibration is aggravated simultaneously; therefore, the overall drawing force goes down.

*3.3. Influence of Drawing Speed on Drawing Force*

Figure 7 displays the influence of wire drawing velocity on the time variation of drawing forces when the distance between the die and reel drum equals half-wavelength and the ultrasonic amplitude remains 10 μm. By comparison, it can be found that the steady-state value of conventional drawing force is not affected by drawing speed, which fluctuates slightly around 22.5 N, because strain rate dependence of the flow stress is not included in the material model. However, the time duration of the elastic deformation stage is prolonged when drawing speed increases, with 1.8 ms for 200 mm/s to 0.5 ms for 800 mm/s. For the ultrasonic drawing stage, the fluctuation range of drawing forces is narrowed with the increment of drawing velocity. And the time consumption for ultrasonic drawing forces to reach the steady value increases correspondingly from 5 oscillation cycles for 200 mm/s to 13 oscillation cycles for 800 mm/s. After the die stops vibrating, the fluctuation of the drawing force continues for a while, but the time duration is shortened with the increment of drawing velocity. According to the hypothesis proposed in Section 3.2, with the increment of drawing speed, the plastic

deformation phase takes up a greater proportion in a vibration period, therefore the overall drawing force is increased. Meanwhile, the stretching vibration time is reduced, more cycles are undergone before the minimum drawing force gets stabilized. In addition, as the drawing speed approaches the threshold value $2\pi fA$, the decrement of minimum drawing force in each cycle drops, making the total drawing force reduction decreased.

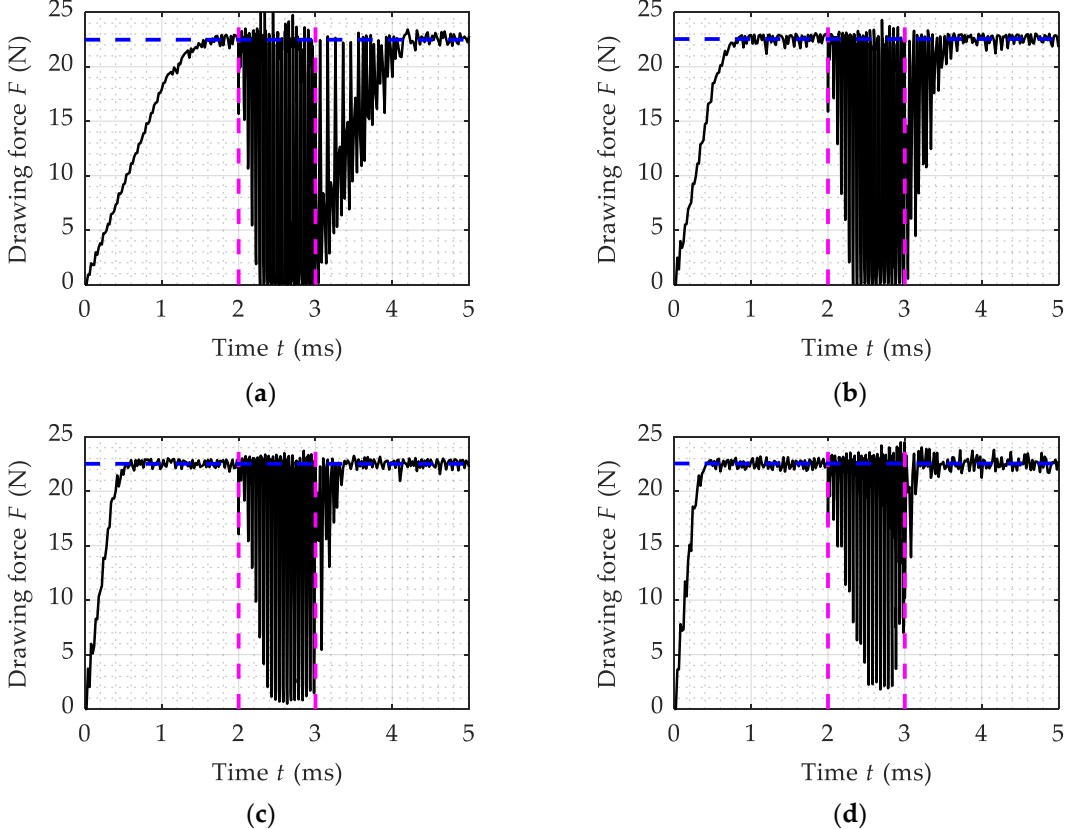

**Figure 7.** Influence of drawing speed on time-variation of the drawing force with reel-die distance of 126.4 mm: (**a**) $V$ = 200 mm/s; (**b**) $V$ = 400 mm/s; (**c**) $V$ = 600 mm/s; (**d**) $V$ = 800 mm/s.

Figure 8 illustrates the time variation of drawing forces at different drawing speed when the die locates 7.9 mm ($\lambda/32$) to the reel drum and the ultrasonic amplitude remains 10 μm. In general, the drawing forces present a similar upward trend as the drawing speed rises. However, the envelope lines of the drawing forces are shaped as rectangular instead of the trapezoid. An abrupt decline in drawing force could be observed in these figures when ultrasonic vibration is imposed. The drawing force will restore to the initial value of 22.5 N immediately when ultrasonic vibrations are removed. Besides, compared with the half-wavelength conditions, the fluctuation range of the corresponding drawing force appears to be more sensitive to drawing speed, which is narrowed down sharply, especially at higher drawing speeds. This phenomenon might be caused by the weakened stretching vibration of uncoiled drawn wire, as the oscillating frequency of the die is far below the first-order longitudinal frequency of the wire.

Figure 9 compares the influence of drawing speed on averaged ultrasonic drawing force when the reel-die distance equals 126.4 mm and 7.9 mm, respectively. Although the two curves present a similar upward trend with the increment of drawing speed, the drawing force at the half-wavelength position is lower than the $\lambda/32$ position. The gap between the two curves is gradually narrowed as the drawing speed rises. When approaching the critical drawing speed 1256 mm/s, which is calculated at the amplitude of 10 μm, and frequency of 20 kHz, the two curves all converge to the conventional drawing force of 22.5 N. The difference between the two curves might be attributed to the stretching

vibration of the wire. At the 7.9 mm condition, drawing force improvement caused by increased drawing velocity could not be compensated by the strengthened oscillation of the wire, as occurs in the half-wavelength condition.

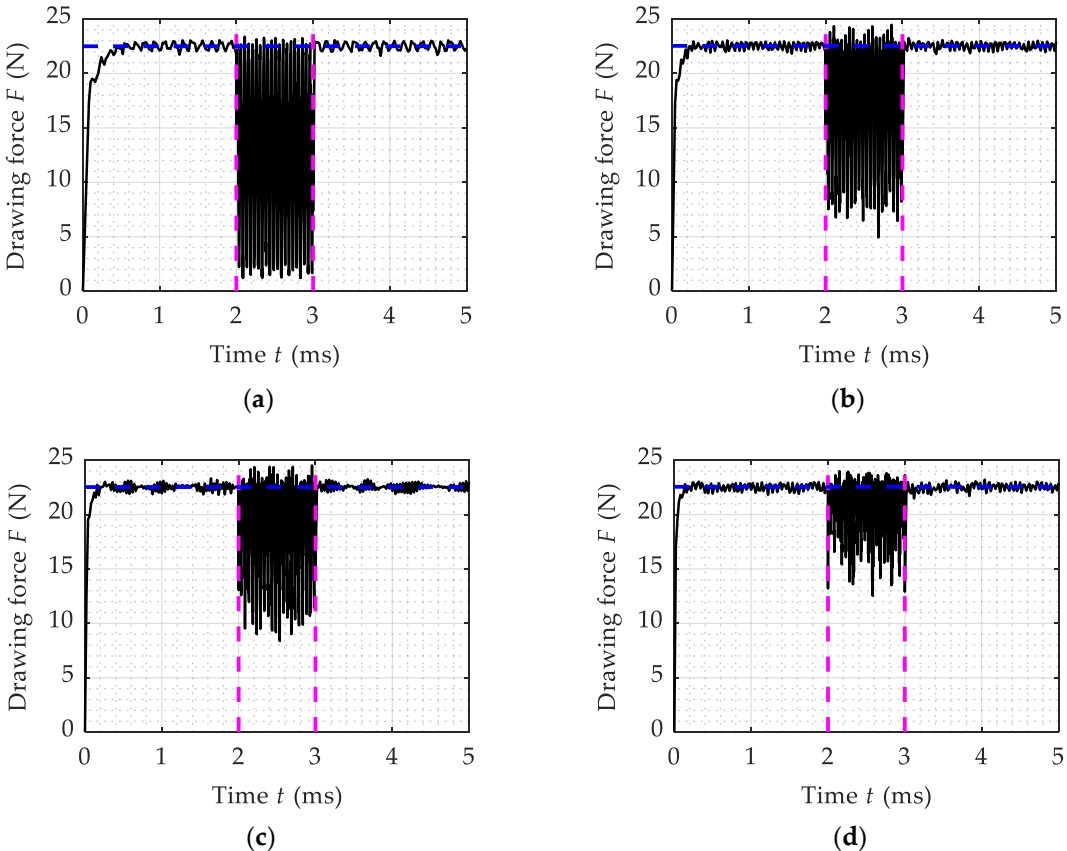

**Figure 8.** Influence of drawing speed on time-variation of the drawing force with reel-die distance of 7.9 mm: (**a**) $V$ = 200 mm/s; (**b**) $V$ = 400 mm/s; (**c**) $V$ = 600 mm/s; (**d**) $V$ = 800 mm/s.

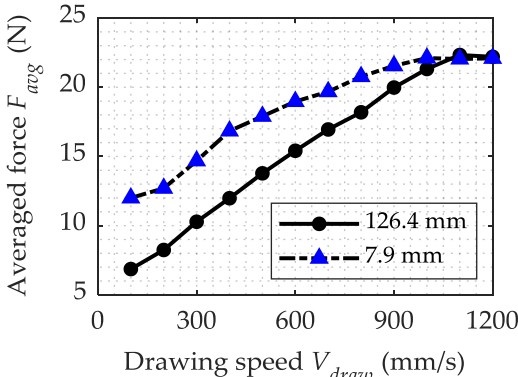

**Figure 9.** FEM calculation results of average ultrasonic drawing force with different drawing speeds with reel-die distance of 126.4 mm and 7.9 mm.

### 3.4. Influence of Friction Coefficient on Drawing Force

To further investigate the influence of friction coefficient on the variation of drawing force, ultrasonic amplitude and drawing speed are specified as 10 µm and 300 mm/s, and remain unchanged. Figure 10 shows the time variation of drawing forces when the reel-die distance equals 15.8 mm ($\lambda$/16) with friction coefficient assigned as 0.3 and 0.5, respectively. Compared with simulation results in

Figures 3 and 4, in the two subfigures of Figure 10, conventional drawing forces climb up to 36.13 N and 45 N from 22.5 N, respectively, and the averaged ultrasonic drawing forces correspondingly ascend to 32.07 N and 41 N from 18.68 N. Therefore, it is suggested that the friction coefficient influences the drawing force no matter whether ultrasonic vibrations are imposed. In addition, the increments of conventional drawing force, when friction coefficient rises, are very close to that of averaged ultrasonic drawing forces. In other words, ultrasonic vibration does not affect the wire-die friction coefficient. However, with excessive friction, plastic deformation would occur to the drawn wire between the die and the reel, resulting in wire breakage, as plotted in Figure 10b.

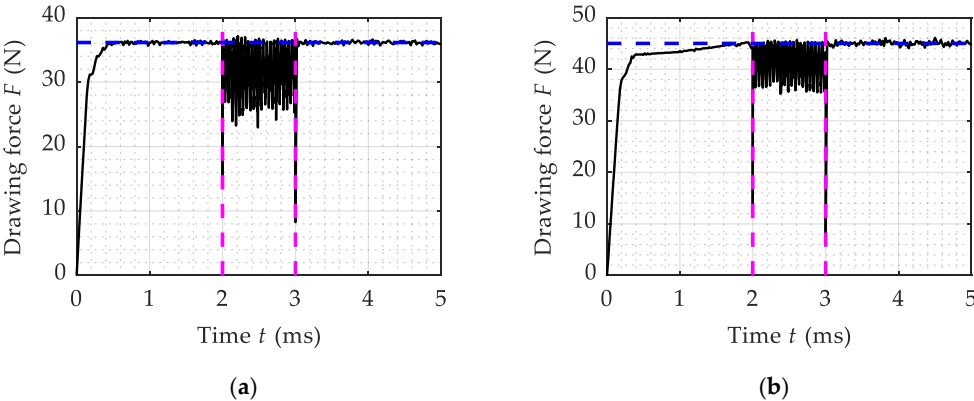

(**a**)          (**b**)

**Figure 10.** Time variation of drawing force with different friction coefficient when the distance between the reel and the die is 15.8 mm: (**a**) $\mu = 0.3$; (**b**) $\mu = 0.5$.

Figure 11 shows the time variation of drawing forces with friction coefficient of 0.3 and 0.5, respectively, when the die locates 126.4 mm from the reel. Conventional drawing forces are 36.1 N and 42.8 N, which are consistent with the results in Figure 10. Averaged ultrasonic drawing forces increase to 17.15 N and 20.83 N from 10.28 N. Similarly, ultrasonic vibration exerts influence on both the conventional and the ultrasonic drawing forces. However, on this occasion, the increments of conventional drawing force caused by increased friction are obviously larger than that of averaged ultrasonic forces. Therefore, at the half-wavelength condition, ultrasonic vibration helps to decrease the equivalent friction with the contacting region. Compared with Figure 3d, severe distortion could be found in the two subfigures. In the conventional drawing phase, time-consuming to reach the steady-state increases to 1.6 ms and 1.8 ms from 1 ms, respectively. This might be caused by the plastic deformation of the uncoiled drawn wire. In the ultrasonic drawing phase, many negative values appear in the drawing force curve. As the flexibility of the titanium is not fully considered in the FE model, the negative section of the curves will be chopped off in the data post-processing procedure.

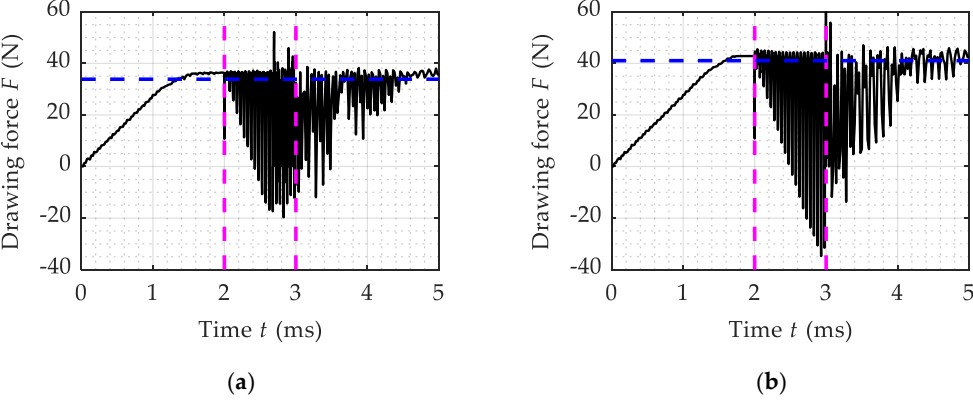

(**a**)          (**b**)

**Figure 11.** Time variation of drawing force with different friction coefficient when the distance between the reel and the die is 126.4 mm: (**a**) $\mu = 0.3$; (**b**) $\mu = 0.5$.

Figure 12 illustrates the influences of friction coefficient on conventional and ultrasonic drawing forces. In both subfigures, the drawing forces ascend with an increased friction coefficient. For conventional drawing, the simulation results are in good agreement with a friction coefficient below 0.4 and are consistent with theoretical results calculated according to Avitzur's theory, when the friction coefficient is lower than 0.2 [23]. For ultrasonic drawing, the drawing force curve has the same shape with the conventional drawing force curve when reel-die distance equals $\lambda/16$. However, at the half-wavelength condition, the increment of ultrasonic drawing force is much smaller than that of conventional drawing force, which means a decreased equivalent friction.

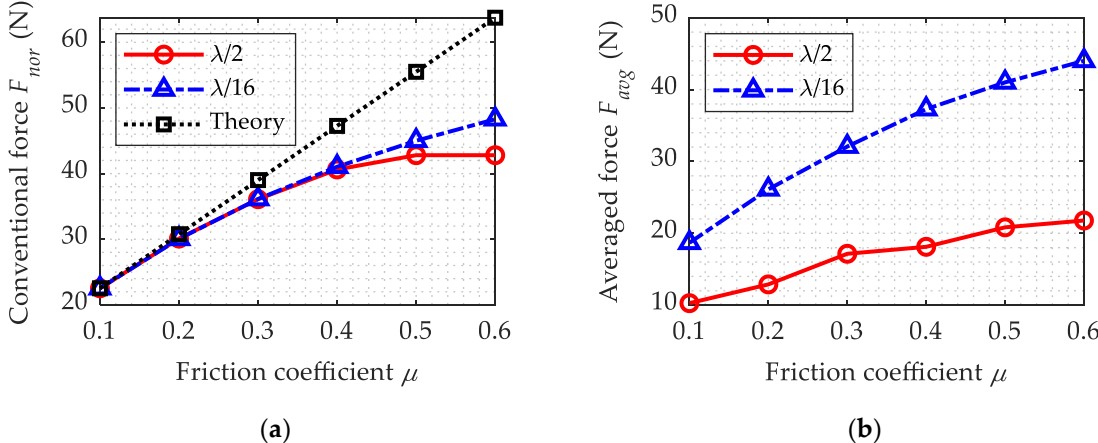

(**a**)　　　　　　　　　　　　　　　(**b**)

**Figure 12.** Influence of friction coefficient on drawing forces when the die locates at different positions: (**a**) Influence of friction on conventional drawing force; (**b**) Influence of friction coefficient on averaged ultrasonic drawing forces.

*3.5. Influence of Ultrasonic Vibration on Stress Distribution*

Figure 13 depicts the Mises equivalent stress distribution of titanium wire under conventional drawing conditions with drawing velocity of 300 mm/s and friction coefficient of 0.1. In the contour, the equivalent stress remains 0~105.9 MPa in a large portion of the feeding area, which locates to the left of the die, when the influence of the back-pull force is neglected. At the entrance region, the Mises stress surged to around 423 MPa. The maximum stress 423.6~529.5 MPa occurs at the reduction area and bearing area, where the die is in direct contact with the wire. However, at the bearing area and the right adjacent area to it, the maximum stress only concentrates on the outer layer of the wire. These are mainly residual stresses caused by the inhomogeneous deformation between the surface and core section of the metallic wire, which declines sharply along the radial direction (−X) of the wire. At the reduction area, however, the equivalent stress maintains 423.6~529.5 MPa throughout the whole internal region of the wire. This is where plastic deformation mainly occurs. For the wires far from the right side of the die, the drawing stress is kept between 211.8 MPa to 311.7 MPa.

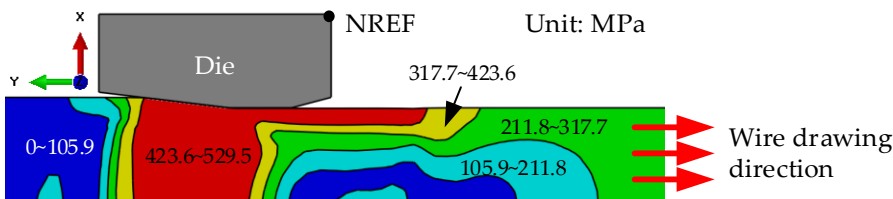

**Figure 13.** Mises equivalent stress distribution of the wire at the contact region under conventional drawing conditions.

To investigate the influence of ultrasonic vibration on the stress distribution of the wire, the simulation is carried out with an ultrasonic amplitude of 10 μm and a wire-die distance of 15.8 mm. The equivalent contours around the contact region for different movement states of the die are demonstrated in Figure 14. It can be seen that, when the die moves, along the reversed wire drawing direction, from the equilibrium position to the leftmost position, the overall stress distribution is similar to that of conventional drawing, except for slight difference at the core segment of the drawn wire, with a maximum stress of 525.9 MPa. In contrast, when the die moves, along the drawing direction, from the central position to the right extreme position, more significant changes could be observed with the maximum equivalent stress dropping to 515 MPa and shrinkage in its area. In addition, the equivalent stress rises to 206 MPa at the feeding area, 308.8 MPa at the core of the drawn wire, and reduced to below 411.8 MPa at the outer surface layer. The more uniform distribution of the equivalent stress is beneficial for eliminating the defects and residual stress caused by inhomogeneous deformation between the core and surface layer of the titanium wire.

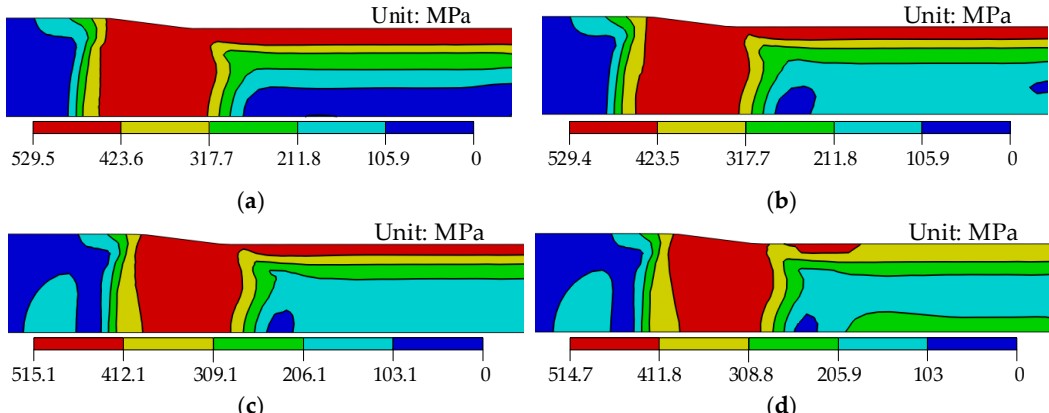

**Figure 14.** Mises equivalent stress distribution of the wire at the contact region under ultrasonic drawing conditions when the die locates 15.8 mm from the reel and in various motion states: (**a**) Equilibrium position and moves to left; (**b**) Left extreme position; (**c**) Equilibrium position and moves to the right; (**d**) Right extreme position.

Figure 15 illustrates the contact and friction stress distribution at the wire-die interface under conventional drawing conditions. From Figure 15a, it can be seen that the maximum value of contact stress occurs at the two sides of the reduction area, adjacent to the entrance and the bearing areas. This is where the surface plastic deformation mainly happens, as the contact stress range, 513.1~641.4 MPa, is apparently above the yield stress of titanium, which is calculated as around 460 MPa based on theoretical and simulation results. In the middle of the reduction area, contact stress falls into the range of 384.8~513.1 MPa, both elastic and plastic deformations coexist. In the entrance and bearing regions, which locate at the two sides of the reduction region, the contact stress plummets to below 128.3 MPa, which means no plastic deformation happens. In Figure 15b, we can find that the friction stress has the same distribution with the contact stress, but is 10% of the latter in value, as the Coulomb friction model is adopted in the setup of the FEA model with a coefficient of 0.1. Therefore, no additional simulation result about the friction stress will be separately presented. In general, the contact and friction stress distribution is compatible with the equivalent stress distribution.

Figure 16 shows the contact stress contours for the ultrasonic drawing process with the die in different motion states. Compared with traditional drawing, there is almost no change in the distribution of contact stress, except for a slight change in the values. When the die locates at the equilibrium position and moves to the left, the maximum contact stress goes up to 660.4 MPa, which is slightly above 641.4 MPa, as shown in Figure 16a. In other positions, the contact stresses are slightly lower than conventional drawing values. In the whole ultrasonic drawing process, the contact stress at the reduction region is apparently above 0. In other words, no separations would occur at the

wire-die interface, as presupposed in the reverse friction mechanism theory. Although with shortened reel-die distance, the minimum contact force could be further decreased, to the point of achieving an intermittent separation between the two parts, which has no practical meaning.

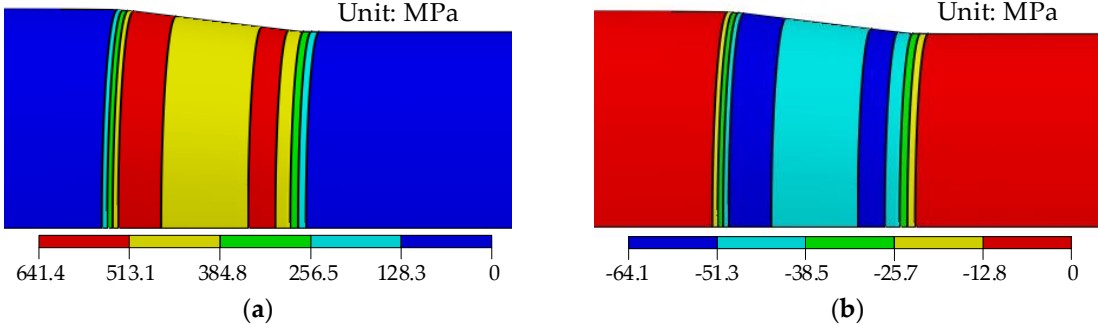

**Figure 15.** Contact stress and friction stress distribution at the contact interface under conventional drawing conditions: (**a**) Contact stress distribution; (**b**) Friction stress distribution.

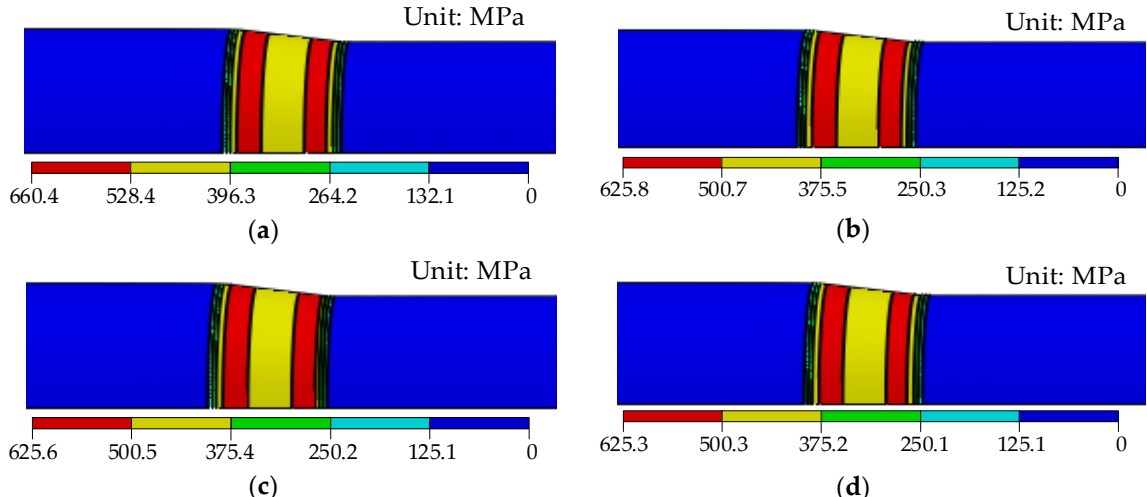

**Figure 16.** Contact stress distribution at contact interface under ultrasonic drawing conditions when the die locates 15.8 mm from the reel and in various motion states: (**a**) Equilibrium position and moves to left; (**b**) Left extreme position; (**c**) Equilibrium position and moves to the right; (**d**) Right extreme position.

## 4. Conclusions

An axisymmetric FE model for a single-pass titanium wire drawing process was built in the commercial software Abaqus based on the practical length of wire. Then, the influence of ultrasonic vibrations on the wire drawing process was investigated with respect to four factors: the location of the die, ultrasonic amplitude, drawing velocity, friction coefficient. At last, the stresses distribution contours at the contact region with and without ultrasonic vibrations were compared. The main conclusions of this study can arrive as follows:

(1) Ultrasonic metal wire drawing process can be interpreted as the frictional contact problem between a certain length of deformable metallic string and the rigid die. The drawing force reduction caused by the addition of ultrasonic vibration could be attributed to the stretching vibration of the wire, which locates between the reel and the die. When the reel-die distance approximates the half-wavelength of ultrasonic vibration when propagating in the metallic material, the drawing force saw the greatest decline.

(2) In the ultrasonic drawing process, an oscillation cycle of the die could be divided into two phases: the plastic deformation phase and the stretching vibration phase. The time distribution between

them is determined by the ultrasonic amplitude and wire drawing speed. The drawing force reduction increases with the increment of ultrasonic amplitude and decrease in drawing speed.

(3) Ultrasonic vibration has almost no effect on the overall friction coefficient, except for the half-wavelength condition or when the reel-die distance is very small.

(4) There is no separation between the wire and the die at the reduction area for single-pass ultrasonic wire drawing when the back-pull force is not considered. Ultrasonic vibrations do have influences on the internal equivalent stress of the wire and the contact stress at the wire-die interface, however, only slightly.

**Author Contributions:** S.L. wrote the paper and completed the numerical simulation; T.X. and X.S. provided the funding and supervised the project; H.C. revised the manuscript and helped in the simulation part. All authors have read and agreed to the published version of the manuscript.

**Funding:** This research was funded by the National Natural Science Foundation of China, grant number 51575130.

**Conflicts of Interest:** The authors declare no conflict of interest.

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
