# Peer review of "Finite Element Analysis on Ultrasonic Drawing Process of Fine Titanium Wire"

_metals, doi:10.3390/met10050575_

Round 1
Reviewer 1 Report
Major revisions:
The following points must be considered before publishing the manuscript.
- What is the novelty the present work? From the mathematical and numerical point of view it seems there is lack of novelty except the change of material.
- The methodology is very similar to previous research works and the significance of results is not well incorporated by comparing it with the previous research to clarify the difference.
- In the article ¨Hayashi, M.; Jin, M.; Thipprakmas, S.; Murakawa, M.; Hung, J.-C.; Tsai, Y.-C.; Hung, C.-H. Simulation of ultrasonic-vibration drawing using the finite element method (FEM). J. Mater. Process. Technol. 2003, 140, 30–35.¨ We can understand in 2003 they did not consider thermo-mechanical behavior but now in 2020 thermo-mechanical models have been developed which should be included as temperature influences the plasticity of material during wire drawing process.
- In the 99 why the generation of heat due to plastic dissipation inside the wire is not considered?
- There is recommendation that authors should consider thermo-mechanical modeling in the simulations to make worth of present work.
Minor revisions:
- In the lines 40, 41 and 42 the reference [8] is for stainless steel wire. Therefore, change it from `` Therefore, ultrasonic drawing is a promising technology to increase productivity, lower the cost, and improve the quality of the products in the industrial manufacturing of titanium wires [8]´´ to ´´ Therefore, ultrasonic drawing is a promising technology to increase productivity, lower the cost, and improve the quality of the products in the wire drawing processes[8].´´
- Change the line 45 from ´´ The idea of introducing ultrasonic vibrations into the drawing process was first reported by Blaha and Langenecker in 1955[12].´´ to ´´ The idea of introducing ultrasonic vibrations into the drawing process was first reported by Blaha et al. [12].
- In the lines 49 to 52 change ´´In 1968, Winsper and Sansome reviewed various experimental results on ultrasonic-assisted metal forming processes and pointed out that all these phenomena could be explained by the superposition of an alternating load on a static load, which is later known as stress superposition hypothesis [13].´´ to ´´Winsper et al. [13] reviewed various experimental results on ultrasonic-assisted metal forming processes and pointed out that all these phenomena could be explained by the superposition of an alternating load on a static load, which is later known as stress superposition hypothesis.´´
- Make similar changes in all the references as explained above.
- In the figures 3,4,5,7,8,10 and 11 the time variation of drawing force was compared with different parameters but the average ultrasonic drawing force should also be considered to make comparison with FEM calculation results of average ultrasonic drawing forces.
- In the figures 6 and 9 change the description like in line 267 ´´ FEM calculation results of average ultrasonic drawing force with different ultrasonic amplitudes at the drawing speeds of 300 mm/s and 600 mm/s.´´
- In the line 217 make correction ´´FEM calculation results of average ultrasonic drawing forces at different distances between the die and the reel drum.´´
- In the lines 74, 75 and 76, make changes from ´´The results indicate that longitudinal vibration is more beneficial for reducing the friction force and improving the surface finish of the drawn wire than the composite vibration.´´ to ´´ The results indicated that longitudinal vibration is more beneficial for reducing the friction force and improving the surface finish of the drawn wire than the composite vibration.´´
- In the lines 78 and 79, make correction from ´´Results show that increased vibration amplitude leads to greater drawing force decrement, whereas increased drawing velocity brings about the reverse effect. ´´to´´ Results showed that increased vibration amplitude leads to greater drawing force decrement, whereas increased drawing velocity brings about the reverse effect.
- In the lines 83-85, make correction from ´´ Present work mainly focusses on experimental research and theoretical studies, whereas little attention was paid to the finite element analysis of the ultrasonic-assisted drawing process.´´ to ´´The recent works mainly focused on experimental research and theoretical studies, whereas little attention was paid to the finite element analysis of the ultrasonic-assisted drawing process.``
- In the line 330 make correction from ´´which is consistent with the results in Figure 10´´ to ´´ which are consistent with the results in Figure 10.´´
Author Response
For reviewer #1:
- What is the novelty the present work? From the mathematical and numerical point of view it seems there is lack of novelty except the change of material.
- The methodology is very similar to previous research works and the significance of results is not well incorporated by comparing it with the previous research to clarify the difference.
Reply: Material is not the point. The work conducted by Hayashi et al. is a specific case covered by our research. Their FEA model is not based on the actual length of the wire, therefore the stretching of the wire is not considered. The rigidity of the wire was exaggerated, which means the equivalent ultrasonic amplitude of the die was unintentionally enlarged. Certainly, some conclusions of them still make sense.
- In the article ¨Hayashi, M.; Jin, M.; Thipprakmas, S.; Murakawa, M.; Hung, J.-C.; Tsai, Y.-C.; Hung, C.-H. Simulation of ultrasonic-vibration drawing using the finite element method (FEM). J. Mater. Process. Technol. 2003, 140, 30–35.¨ We can understand in 2003 they did not consider thermo-mechanical behavior but now in 2020 thermo-mechanical models have been developed which should be included as temperature influences the plasticity of material during wire drawing process.
- In the 99 why the generation of heat due to plastic dissipation inside the wire is not considered?
- There is recommendation that authors should consider thermo-mechanical modeling in the simulations to make worth of present work.
Reply: Neglect of thermal factor will not affect the correctness of the general conclusion in our research. Our manuscript focuses on manifesting the generalized conclusion of how ultrasonic vibration works in fine metallic wire drawing process. Even if the material is changed, the conclusion still holds. In addition, thermo-mechanical theory has been fully developed decades ago, and could be employed in commercial software before 2003. There are always simplifications in simulation, and we have to differentiate the importance of these factors.
- In the lines 40, 41 and 42 the reference [8] is for stainless steel wire. Therefore, change it from `` Therefore, ultrasonic drawing is a promising technology to increase productivity, lower the cost, and improve the quality of the products in the industrial manufacturing of titanium wires [8]´´ to ´´ Therefore, ultrasonic drawing is a promising technology to increase productivity, lower the cost, and improve the quality of the products in the wire drawing processes[8].´´
- Change the line 45 from ´´ The idea of introducing ultrasonic vibrations into the drawing process was first reported by Blaha and Langenecker in 1955[12].´´ to ´´ The idea of introducing ultrasonic vibrations into the drawing process was first reported by Blaha et al. [12].
- In the lines 49 to 52 change ´´In 1968, Winsper and Sansome reviewed various experimental results on ultrasonic-assisted metal forming processes and pointed out that all these phenomena could be explained by the superposition of an alternating load on a static load, which is later known as stress superposition hypothesis [13].´´ to ´´Winsper et al. [13]reviewed various experimental results on ultrasonic-assisted metal forming processes and pointed out that all these phenomena could be explained by the superposition of an alternating load on a static load, which is later known as stress superposition hypothesis.´´
- Make similar changes in all the references as explained above.
- In the figures 6 and 9 change the description like in line 267 ´´ FEM calculation results of average ultrasonic drawing force with different ultrasonic amplitudes at the drawing speeds of 300 mm/s and 600 mm/s.´´
- In the line 217 make correction ´´FEM calculation results of averageultrasonic drawing forces atdifferent distances between the die and the reel drum.´´
- In the lines 74, 75 and 76, make changes from ´´The results indicate that longitudinal vibration is more beneficial for reducing the friction force and improving the surface finish of the drawn wire than the composite vibration.´´ to ´´ The resultsindicated that longitudinal vibration is more beneficial for reducing the friction force and improving the surface finish of the drawn wire than the composite vibration.´´
- In the lines 78 and 79, make correction from ´´Results show that increased vibration amplitude leads to greater drawing force decrement, whereas increased drawing velocity brings about the reverse effect. ´´to´´ Results showedthat increased vibration amplitude leads to greater drawing force decrement, whereas increased drawing velocity brings about the reverse effect.
- In the lines 83-85, make correction from ´´ Present work mainly focusses on experimental research and theoretical studies, whereas little attention was paid to the finite element analysis of the ultrasonic-assisted drawing process.´´ to ´´The recent works mainly focusedon experimental research and theoretical studies, whereas little attention was paid to the finite element analysis of the ultrasonic-assisted drawing process.``
- In the line 330 make correction from ´´which is consistent with the results in Figure 10´´ to ´´ which areconsistent with the results in Figure 10.´´
Reply: Above errors have been revised as suggested.
- In the figures 3,4,5,7,8,10 and 11 the time variation of drawing force was compared with different parameters but the average ultrasonic drawing forceshould also be considered to make comparison with FEM calculation results of average ultrasonic drawing forces.
Reply: Figures 3,4,5,7,8,10 and 11 are simulation result of drawing forces, which includes both conventional drawing force and ultrasonic drawing force. They are real-time values. Figure 6, 9, 12, are about calculated averaged values of ultrasonic drawing forces, which are based on the 2~3 ms sections of figures 3,4,5,7,8,10 and 11. Each data point of the former corresponds to a subfigure of the latter. So, compassions have been taken into account in figure 6, 9, 12.
Reviewer 2 Report
The proposed paper is focused on the effect of the ultrasound implementation in the wiredrawing process. The results are based on results obtained by finite element method determinating the influence of this additional effect in the drawing force, discusing it in terms of different values for the vibration amplitude, the drawing speed, the distance between the die and the traction reel and the friction ratio. The numerical results show that a higuer amplitude trhows, in general, a significative reduction in the drawing force required.
I propose to apply some improvements to the work what happened to list below:
1.- Please, rewritte the sentence "..., the drawing force decrement is maximized when the die locates......, while minimized at the quarter wavelenght position." (lines 22-23)
as follows: "..., the drawing force is minimized when the die locates......, while maximized at the quarter wavelenght position."
2.- In the lines 98-99, correct the simbol Ø (typing Alt+0216) or write "0.4 millimeters diameter" or "a diameter of 0.4 illimeters".
3.- Line 99, The term "reduction is redundant". Plaease, correct it. Read all the texts carefully and correct this type of issues.
4.- Figure 1, In the right (general view) of the graph the drawing die is very small and invisible in this scale. It is advisable to frame the point where the drawing die is (PP1) and indicate with an arrow towards the left graphic (enlarged), indicating to the left graphic as an enlargement of the framed area.
5.- Lines 131-134, correct the term "Passion" as "Poisson"
6.- The material normalized definition TA2 must to be justified by the corresponding citation to the Norm considered and this Standard norm must to be included as a reference.
7.- Please, the meaning of the sentence "As the drawn wire performs forced vibration under the simulate of the die, simulations are conducted under the occasion.... respectivelly,...) is not clear. You must rethink and rewrite this phrase for a better understanding. (lines 170-172)
8.- The language grammar and spelling must to be checked carefully since there are some paragraphs containing sentences and terms that must to be simplified or improbed for a beter comprehension by the reader.
9.- Line 224, you wrote: "From previous studies, ... Please, cite these works.
10.- Figure 12, please correct the text "Theretical" in the graph legend.
11.- Figure 13, please correct the text "dierction" in the graph text.
12.- Figures 13, 14, 15, 16; please, Extend a little more the distance between the previous paragraph and the figure in each case.
In all these figures except the Figure 13, please unify the form in wich the Unit: MPa is presented (above the wire or at the right side of the stress colored scale).
13.- The conclusions of the work are based solely on results obtained from numerical simulations that are not experimentally contrasted. It would be very convenient and interesting to include a section for validation of the results, for example by comparing the value of the drawing force obtained in the FEM software with that obtained in a real validation test. In this way the validity of the simulations would be confirmed.
I reccomend, at least, explain in detail how mechanical tests have been performed to determine the parameters that define the Multi-Linear Kinematic Hardening (MKINH) strain hardening behavior model (nr of specimens/tests, dimensions of specimens, material parameters associated to the used model,etc).
Despite this last point, the work seemed very interesting to me.
Author Response
For reviewer #2:
1.- Please, rewritte the sentence "..., the drawing force decrement is maximized when the die locates......, while minimized at the quarter wavelenght position." (lines 22-23) as follows: "..., the drawing force is minimized when the die locates......, while maximized at the quarter wavelenght position."
2.- In the lines 98-99, correct the simbol Ø (typing Alt+0216) or write "0.4 millimeters diameter" or "a diameter of 0.4 illimeters".
3.- Line 99, The term "reduction is redundant". Plaease, correct it. Read all the texts carefully and correct this type of issues.
4.- Figure 1, In the right (general view) of the graph the drawing die is very small and invisible in this scale. It is advisable to frame the point where the drawing die is (PP1) and indicate with an arrow towards the left graphic (enlarged), indicating to the left graphic as an enlargement of the framed area.
5.- Lines 131-134, correct the term "Passion" as "Poisson"
7.- Please, the meaning of the sentence "As the drawn wire performs forced vibration under the simulate of the die, simulations are conducted under the occasion.... respectivelly,...) is not clear. You must rethink and rewrite this phrase for a better understanding. (lines 170-172)
8.- The language grammar and spelling must to be checked carefully since there are some paragraphs containing sentences and terms that must to be simplified or improbed for a beter comprehension by the reader.
9.- Line 224, you wrote: "From previous studies, ... Please, cite these works.
10.- Figure 12, please correct the text "Theretical" in the graph legend.
11.- Figure 13, please correct the text "dierction" in the graph text.
12.- Figures 13, 14, 15, 16; please, Extend a little more the distance between the previous paragraph and the figure in each case.
In all these figures except the Figure 13, please unify the form in wich the Unit: MPa is presented (above the wire or at the right side of the stress colored scale).
Reply: Above problems have been revised as suggested.
6.- The material normalized definition TA2 must to be justified by the corresponding citation to the Norm considered and this Standard norm must to be included as a reference.
Reply: TA2 is a well-known grade of commercial pure titanium. I do not think it is necessary to add a specific. The core of this manuscript is how ultrasonic vibration works in the fine metallic wire drawing process. The conclusion can be generalized to other metals. TA2 is just a study case. The chemical composition can be referenced in the following website, if interested: https://www.aircraftmaterials.com/data/titanium/ta2.html
12.- Figures 13, 14, 15, 16; please, Extend a little more the distance between the previous paragraph and the figure in each case.
Reply: If so, there will be blanks between pages. I believe the editor will deal with the problems.
13.- The conclusions of the work are based solely on results obtained from numerical simulations that are not experimentally contrasted. It would be very convenient and interesting to include a section for validation of the results, for example by comparing the value of the drawing force obtained in the FEM software with that obtained in a real validation test. In this way the validity of the simulations would be confirmed.
Reply:We have published several papers about the experiment aspect of metallic drawing. But the mechanism behind is not fully explained. For TA2 wire with a diameter of 0.4 mm, you can refer to citations [19-20], and the following paper for experimental parameters. Experimental Study on Titanium Wire Drawing with Ultrasonic Vibration, Ultrasonics (ISSN:0041-624X). 2018, 83: 60-67.
Thank you for your revision suggestions, which really helps for enhancing the quality of this manuscript.
Reviewer 3 Report
The paper proposal presents a FEM parametric study dealing with ultrasonic drawing of Titanium wire. The effect (on the drawing process) of the distance between the reel and the die, (die) vibration amplitude, drawing velocity and coefficient of friction is studied.
Although the results can be interesting, the scientific importance of the whole work is, however, quite limited.
The major weakness consists in the missing comparison with the real (experimental) process characteristics, i.e. no computed value (e.g. of average drawing force) was compared with experimentally obtained one. This is a very important point which must be improved (this fact, in my opinion, impedes the publication of the paper at the present state).
English need tidying.
Some other remarks:
Correct nomenclature should be used – e.g.:
The (die) vibration amplitude is inaccurately denoted as ultrasonic amplitude (throughout the text).
Coefficient of friction is wrongly denoted as friction ratio (throughout the text).
Line 99-101 The statement “The generation of heat due to plastic dissipation inside the wire and frictional heat generation within the contact region are not considered.” need to be discussed (and/or explained why it is not considered). The mechanical response can be strongly sensitive to temperature as well as to the strain rate (which is also not considered in the paper).
Line 202 What authors mean by the statement “without the consideration of the tenderness of the metallic wire” ?
Line 243 Should probably be “A=8 μm” instead “L=8 μm”
In the whole paragraph 3.3. Influence of Drawing Speed on Drawing Force the ultrasonic amplitude is not specified. Is it 10 μm?
In my opinion, the paper should be completed by the comparison with the experiment so the results could be discussed in a more comprehensive way (before the paper could be recommended for the publication).
Author Response
For reviewer #3:
Correct nomenclature should be used – e.g.:
The (die) vibration amplitude is inaccurately denoted as ultrasonic amplitude (throughout the text).
Coefficient of friction is wrongly denoted as friction ratio (throughout the text).
Reply: In this manuscript, ultrasonic amplitude denotes the ultrasonic amplitude of the die. And the friction ratio is a synonymy for coefficient of friction. These are common used in research papers and we do not think they are confusing.
Line 99-101 The statement “The generation of heat due to plastic dissipation inside the wire and frictional heat generation within the contact region are not considered.” need to be discussed (and/or explained why it is not considered). The mechanical response can be strongly sensitive to temperature as well as to the strain rate (which is also not considered in the paper).
Reply: There are many influential factors for a real metal-forming process, which have to be trade-off in the numerical simulation process. The core of this manuscript is to investigate the influence of ultrasonic vibrations on the metallic wire drawing process. The conclusion is general, not limited to titanium wire. The influence of heat generation and rate-dependence are interesting, but not within the scope of this manuscript. Hope you understand.
Line 202 What authors mean by the statement “without the consideration of the tenderness of the metallic wire” ?
Reply: For a long and thin wire, it can bear the tension force but not the pressure. The tension force on the string should be above 0, otherwise, it will not be kept straight.
Line 243 Should probably be “A=8 μm” instead “L=8 μm”
Reply: The error has been revised as suggested. Thank you.
In the whole paragraph 3.3. Influence of Drawing Speed on Drawing Force the ultrasonic amplitude is not specified. Is it 10 μm?
Reply: Yes, the amplitude is kept as constant 10 μm. Revisions have been made.
The major weakness consists in the missing comparison with the real (experimental) process characteristics, i.e. no computed value (e.g. of average drawing force) was compared with experimentally obtained one. This is a very important point which must be improved (this fact, in my opinion, impedes the publication of the paper at the present state).
In my opinion, the paper should be completed by the comparison with the experiment so the results could be discussed in a more comprehensive way (before the paper could be recommended for the publication).
Reply: This manuscript is just about the numerical simulation aspect of ultrasonic-assisted drawing process of fine metallic wires, which is targeted at the Special Issue: Numerical Modelling and Simulation of Metal Processing. We believe the manuscript is within the scope. As for the experimental aspect, you can refer to citations [19-20], and the following paper, which share the same experimental instruments and material parameters with this manuscript: Experimental Study on Titanium Wire Drawing with Ultrasonic Vibration, Ultrasonics (ISSN:0041-624X). 2018, 83: 60-67.
English need tidying.
Reply: We have re-read the manuscript and make some revisions.
Reviewer 4 Report
Ultrasonic Drawing is gaining importance because - as the authors emphasized - it allows obtaining material with very good properties, and in addition this method is environmentally friendly and cheaper than the traditional one.
I read the article presented by the authors with curiosity and I must admit that it is well written, thought out, the theoretical part correctly introduces the reader to the topic, and the articles to which the authors refer are selected correctly. The authors focused in this work on the numerical analysis of the process and conducted it efficiently, demonstrating knowledge and skilful conduct of the study, which showed the impact of four factors: the location of the die, ultrasonic amplitude, drawing velocity, and friction ratio on the wire drawing process.
There was no experimental part in this work, but the authors referred to another article in this work, which they already published and which presented the experiment, so the numerical study seems to be a logical element of the whole project.
In my opinion, the article is properly constructed, it is easy to read. Both the assumptions and the numerical study are clearly described, and the conclusions presented in the summary are correctly formulated and clearly indicate that the topic of ultrasonic drawing is current and can be further developed.
Author Response
For reviewer #4:
Ultrasonic Drawing is gaining importance because - as the authors emphasized - it allows obtaining material with very good properties, and in addition this method is environmentally friendly and cheaper than the traditional one.
I read the article presented by the authors with curiosity and I must admit that it is well written, thought out, the theoretical part correctly introduces the reader to the topic, and the articles to which the authors refer are selected correctly. The authors focused in this work on the numerical analysis of the process and conducted it efficiently, demonstrating knowledge and skilful conduct of the study, which showed the impact of four factors: the location of the die, ultrasonic amplitude, drawing velocity, and friction ratio on the wire drawing process.
There was no experimental part in this work, but the authors referred to another article in this work, which they already published and which presented the experiment, so the numerical study seems to be a logical element of the whole project.
In my opinion, the article is properly constructed, it is easy to read. Both the assumptions and the numerical study are clearly described, and the conclusions presented in the summary are correctly formulated and clearly indicate that the topic of ultrasonic drawing is current and can be further developed.
Reply: Thank you for your positive comments and preference of our manuscript, which will surely encourage us to go further in this area.
Round 2
Reviewer 1 Report
In my opinion although the results are in agreement with the previous research but to make the manuscript scientifically sound, some recommendations must be taken into account before publication.
In the line 99, ¨the generation of heat due to plastic deformation is not considered¨. The generation of heat should be considered by developing plasticity model in simulations to study the effect of temperature and plastic strains on mechanical and electroplastic behavior of material.
In the figures 3,4,5,7,8,10 and 11, the time variation of drawing force was compared with different parameters but the experimental average values of ultrasonic drawing forces should be compared with the calculated average values of ultrasonic drawing forces to create better understanding of manuscript.
It is suggested to use one term either ¨friction coefficient¨ or ¨friction ratio¨ throughout the text to avoid any confusion.
Author Response
In the line 99, ¨the generation of heat due to plastic deformation is not considered¨. The generation of heat should be considered by developing plasticity model in simulations to study the effect of temperature and plastic strains on mechanical and electroplastic behavior of material.
Reply: Heat generation and many other factors may also exert influence to the metal forming process, whether for the conventional drawing process or for the ultrasonic drawing process. However, it is not the research point of this paper. There are various specialized and detailed papers for the reader’s reference for each factor in the metal-forming area. No paper covers all these factors. This paper only deals with the ‘ultrasonic’ caused problem by numerical simulation. We will emphasize this point in the manuscript where relevant description first appears.(Line 100-101 after revision). Our finding is that the reduction in drawing force results from two factors: the stretching vibration of the wire between the die and the reel, and the time distribution between the plastic deformation phase and the stretching vibration phase in a motion cycle of the die. Neither of the them is influence by heat generation. Even when the material itself has been changed, above conclusion still make sense, as the phenomenon is independently caused by the addition of ultrasound. What we talk about in this paper is how ultrasound works in the drawing process of long metallic wires with a small cross section. Hope you can understand.
In the figures 3,4,5,7,8,10 and 11, the time variation of drawing force was compared with different parameters but the experimental average values of ultrasonic drawing forces should be compared with the calculated average values of ultrasonic drawing forces to create better understanding of manuscript.
Reply: As the article “Hayashi, M.; Jin, M.; Thipprakmas, S.; Murakawa, M.; Hung, J.-C.; Tsai, Y.-C.; Hung, C.-H. Simulation of ultrasonic-vibration drawing using the finite element method (FEM). J. Mater. Process. Technol. 2003, 140, 30”, which you have mentioned in the first-round revision, this paper is also a pure numerical study with the thermo factors excluded. And this manuscript is submitted to the special issue ’Numerical Modelling and Simulation of Metal Processing’. We believe its content is well within the scope. You can check the scope in the following website: https://www.mdpi.com/journal/metals/special_issues/modelling_simulation_metal_processing
Several papers regarding pure numerical simulation for other manufacturing process have been published in the special issue, which are open access and can be found at the bottom sections of the page.
It is suggested to use one term either ¨friction coefficient¨ or ¨friction ratio¨ throughout the text to avoid any confusion.
Reply: Above question has been revised as suggested.
Although with different opinions, we still thank you for your advice, which encourage us to rethink the highlights of this paper.
Reviewer 2 Report
Dear Authors,
I think that your revision is raseonably agree with my last suggestion.
There is "passion" again in your work (line 134), please correct it.
Another little mistake is the that the position of "(a)" is not in the same line with "(b)", in the Figure 12. Revise and correct please.
Continue working with passion. Congratulatios for your work.
Author Response
I think that your revision is raseonably agree with my last suggestion.
There is "passion" again in your work (line 134), please correct it.
Another little mistake is the that the position of "(a)" is not in the same line with "(b)", in the Figure 12. Revise and correct please.
Continue working with passion. Congratulatios for your work
Reply: The two errors have been revised as suggested. Thank you for your humorous comments. They really contribute to improving the readability and quality of this manuscript.
Reviewer 3 Report
Correct nomenclature should be used – e.g.:
Coefficient of friction is wrongly denoted as friction ratio (throughout the text). Friction ratio is something different than the coefficient of friction.
No comparison with the real (experimental) process characteristics are in the paper. At least a comment should be there (otherwise the impact of FEM modelling is very limited).
Line 99-101 The statement “The generation of heat due to plastic dissipation inside the wire and frictional heat generation within the contact region are not considered.” need to be discussed (and/or explained why it is not considered). The mechanical response can be strongly sensitive to temperature as well as to the strain rate (which is also not considered in the paper). At least a comment should be there.
Line 202 What authors mean by the statement “without the consideration of the tenderness of the metallic wire” ?
Some corrections have to be made before the paper can be recommended for publication (I do not recommend the paper for publication in the present state).
Author Response
Coefficient of friction is wrongly denoted as friction ratio (throughout the text). Friction ratio is something different than the coefficient of friction.
Reply: Yes, indeed we are wrong. The ‘friction ratio’ has been revised as ‘friction coefficient’ or ‘friction factor’ instead. Thank you for pointing it out again.
No comparison with the real (experimental) process characteristics are in the paper. At least a comment should be there (otherwise the impact of FEM modelling is very limited).
Line 99-101 The statement “The generation of heat due to plastic dissipation inside the wire and frictional heat generation within the contact region are not considered.” need to be discussed (and/or explained why it is not considered). The mechanical response can be strongly sensitive to temperature as well as to the strain rate (which is also not considered in the paper). At least a comment should be there.
Reply: This manuscript focuses on the general ultrasonic drawing process of long thin metallic wires. It demonstrates how ultrasound works in this general metal forming process. And these effects or phenomenon caused by ultrasound are independent, which have nothing to do with the heat generation. Even when the elastic-plastic curve or the material itself has been changed, the way ultrasonic works will not change. What the manuscript intends to convey is the general conclusion of how ultrasonic vibration functions in the drawing process of general metallic wires with small cross-sections, instead of a specific experimental process. In that case, the manuscript would be nothing but a case study, which is not our purpose. Certainly, the parameters are based on a real experimental process, which has been cited in the manuscript, as we have replied previously.
For the influence of heat generation, there are many papers specialized in this topic for the reader’s reference. However, it is not the research point of this manuscript.
This manuscript is a numerical study, which is submitted to the special issue of ’Numerical Modelling and Simulation of Metal Processing’. We believe its content is well within the scope.
Line 202 What authors mean by the statement “without the consideration of the tenderness of the metallic wire” ?
Reply: As we have previously replied, a long thin wire could not bear the pressure. Therefore, the real tension should never be below zero. Please notice that it is a ‘wire’ instead of a ‘bar’. We believe we have explained it as simple as possible.
Despite with different understandings, we still thank you for your advice, which encourage us to rethink the emphasis of this paper.
Round 3
Reviewer 3 Report
The answer to the reviewers’ comments should be included in the text.
Author Response
The answer to the reviewers’ comments should be included in the text.
Reply: Above question has been revised as suggested. Thank you for all your advice, which will make the paper easier to understand.